# Prevalence of fungal infections in a patient cohort in The Gambia: Identification and characterization of three priority fungal species in patients with symptoms suggestive of TB

Zackary Salem-Bango[1,2]*, Mary Basiru Njai[3], Zainab Tambajang[3], Tessa Rose Cornell[4], Jayne Sutherland[5], Karen Forrest[6], Omai Garner[1], Behzad Nadjm[6], Saffiatou Darboe[7], Basil Sambou[7], Sheikh Jarju[3]

1 Department of Pathology and Laboratory Medicine, David Geffen School of Medicine, University of California Los Angeles, Los Angeles, California, United States of America, 2 Department of Emergency Medicine, University of Washington, Seattle, Washington, United States of America, 3 Molecular Diagnostics Laboratory, MRC Unit The Gambia at LSHTM, Fajara, The Gambia, 4 Institute of Infection, Veterinary and Ecological Sciences, University of Liverpool, Liverpool, United Kingdom, 5 Vaccines and Immunity Theme, MRC Unit The Gambia at LSHTM, Fajara, The Gambia, 6 Clinical Services Department, MRC Unit The Gambia at LSHTM, Fajara, The Gambia, 7 Tuberculosis Laboratory, MRC Unit The Gambia at LSHTM, Fajara, The Gambia

* zsbango@uw.edu

## Abstract

Invasive fungal infections are an increasing global health concern, particularly in low-resource settings where diagnostic capacity is limited. In The Gambia, where tuberculosis is highly prevalent, fungal infections may be misdiagnosed as mycobacterial disease due to overlapping clinical symptoms and limited access to fungal testing. This study aimed to determine the presence of three priority fungal pathogens by the World Health Organization (WHO) classification system—*Aspergillus* species (*spp.*), *Histoplasma spp.*, and *Pneumocystis jirovecii*—in patients with symptoms suggestive of tuberculosis, and to develop a molecular tool to support future surveillance. A multiplex quantitative polymerase chain reaction assay was developed and validated for simultaneous detection of *Aspergillus spp.*, *Histoplasma spp.*, and *Pneumocystis jirovecii* DNA in human sputum. The assay was applied to 273 stored sputum samples collected from adult patients presenting with respiratory symptoms concerning for tuberculosis in The Gambia. The multiplex assay demonstrated high sensitivity and specificity, detecting as few as ten DNA copies per reaction for each target. Among the 273 sputum samples analyzed, *Aspergillus* DNA was identified in five samples (1.8%), *Pneumocystis jirovecii* DNA in three samples (1.1%), and no samples were positive for *Histoplasma*. Cough and weight loss were the most frequently reported symptoms among participants with positive results. This study represents the first molecular detection of *Aspergillus spp.* and *Pneumocystis jirovecii* in adults in The Gambia. These findings suggest that fungal colonization or infection may occur in a small proportion of patients presenting with tuberculosis-like symptoms. The

**Data availability statement:** Patient level data are available from The Gambia Government/ MRC Joint Ethics Committee for researchers who meet the criteria for access to confidential data. This data is stored within a local BioBank. Otherwise, all relevant data are within the manuscript. Please contact information@mrc. gm for more information or to apply to access the primary data for this study. MRC Unit The Gambia at LSHTM Atlantic Boulevard Fajara PO Box 273 Banjul The Gambia Telephone: (+220) (0) 4495835 / 4495443-6.

Research reported in this publication was supported by the Global Health Program within the David Geffen School of Medicine at University of California Los Angeles. The funders had no role in study design, data collection and analysis, decision to publish, or preparation of the manuscript.

**Competing interests:** The authors have declared that no competing interests exist.

multiplex molecular platform developed here provides an accessible approach for fungal surveillance and may improve diagnosis and management of fungal infections in resource-limited settings.

## Introduction

The 2022 "WHO fungal priority pathogens list to guide research, development and public health action" highlighted the underrecognized and increasing global health threat of invasive fungal disease [1]. This comprehensive guide classified nineteen fungal pathogens as critical, high and medium priority. Prioritization criteria included trends in incidence and prevalence, rate or level of antifungal resistance, and access to diagnostics and evidence-based treatment options. An estimated 6.5 million cases of invasive fungal infections occur worldwide annually: leading to 3.8 million deaths, of which 2.5 million are directly attributable to fungal infections [2]. This underscores the critical need for improved surveillance and diagnostic and treatment modalities to address the global burden of invasive fungal disease.

In underrepresented and under resourced settings worldwide, including The Gambia, the limited availability of fungal diagnostics and surveillance tools in reference or institutional laboratories creates data gaps which prevent accurate estimation of the global fungal disease burden [3]. Conventional gold-standard approaches for mycology diagnostics include direct microscopy, histopathology, culture, and antigen detection [4]. However, despite the availability and feasibility of culture and microscopy techniques, they present significant limitations including long growth time and low sensitivity. Although standard fungal media culture can be accessible and cost-effective for some low-resource settings, many target species/clades require special media [5] and skilled personnel for accurate pathogen identification, and contamination is common. Other techniques, such as antigen-based detection, is prohibitively costly when needing to screen a single patient for multiple fungal targets in low-resource settings or when a large cohort/population needs to be screened.

Molecular-based assays like quantitative Polymerase Chain Reaction (qPCR) offer high sensitivity and specificity and relatively short turn-around time, making them ideal for surveillance or diagnostic efforts in certain settings equipped with the necessary tools [6]. Additionally, once staff are trained on a single assay, it is relatively easy to cross-train technicians and implement novel tests. Furthermore, utilization of multiplex qPCR assays allows the simultaneous identification of multiple pathogens in a single reaction, effectively reducing diagnostic delays and associated per-reaction costs if assays are designed appropriately [7,8].

In The Gambia, there is a significant lack of data on fungal disease burden, likely due to misdiagnosis and underreporting. Additionally, because antifungals are prohibitively expensive and mostly inaccessible in this setting, there has been little work to investigate, classify, and diagnose these pathogens. These limitations present a major challenge in the surveillance and management of infections [3,9,10]. Previous studies in other west African nations have provided preliminary evidence of high rates

of fungal infections among immunocompromised individuals [10]. Furthermore, secondary to high concurrent rates of tuberculosis (TB), fungal infections are often at risk of being misdiagnosed as mycobacterial processes, such as has been documented for *Histoplasma spp.* and *Aspergillus spp.*[11,12]. The incidence rate of TB in The Gambia is 142 per 100,000 [13], which is relatively high. Additionally, many fungal species named in the WHO priority pathogen list [1] remain largely uncharacterized in this setting. Three of these fungi were investigated in this study.

*Aspergillus* spp. is one of the most widespread airborne saprophytic fungi that can lead to disease in humans [14]. Immunocompetent hosts rarely experience symptoms related to infection due to rapid elimination of conidia, regardless of the causative species. In Africa, chronic aspergillosis has been well established as a sequela to the high burden of pulmonary TB on the continent [14,15]. A recent study in Sierra Leone, a TB-endemic country, showed that 20.8% of patients with chronic respiratory symptoms were seropositive for *Aspergillus* spp. and 11.6% had chronic pulmonary aspergillosis, highlighting the significance of establishing fungal infection prevalence in these regions [16]. However, there have been no publications to date on *Aspergillus* spp. in The Gambia.

*Histoplasma* spp. are dimorphic fungi and are the causative agent of histoplasmosis, associated with tropical and sub-tropical climates (which are conducive to growth and proliferation of the saprophytic mycelial form [17]). Data on histoplasmosis burden, particularly the pulmonary form, are sparse in The Gambia. One historical case report [9,18] on histoplasmosis, describing cutaneous and bone lesions, has been documented but lacked epidemiological data to establish the true burden of and associated risk factors for disease. More recent efforts towards addressing this evidence gap estimated *Histoplasma spp.* seroprevalence to be 28.8% in active TB patients and 18.8% in a general population in The Gambia [19].

*Pneumocystis jirovecii* is an opportunistic fungi best known for causing pulmonary disease in immunocompromised individuals with underlying conditions such as human immunodeficiency virus (HIV) [20]. Like *Aspergillus* spp., colonization relates to the host's immune status. Depending on the severity of infection, patients present with cough, dyspnea, and in rare instances (less common than aspergillosis) hemoptysis and respiratory failure [21]. A large multicenter study in The Gambia on childhood pneumonia identified 31 cases in a pediatric sub-population [22]. Besides this study, *Pneumocystis jirovecii* remains completely uncharacterized in adults in The Gambia.

This study aimed to: (i) develop a multiplex qPCR assay to detect *Aspergillus* spp., *Histoplasma* spp. and *Pneumocystis jirovecii* for use on human sputum samples; and (ii) apply the qPCR to examine the burden of fungal DNA in sputum samples from patients with symptoms suggestive of pulmonary tuberculosis in The Gambia.

## Methods

### Sample selection, collection and storage

273 deidentified sputum samples, collected between February 2017 and April 2021, were analyzed between February 1st and June 1st, 2024 at the Medical Research Council (MRC) Unit, The Gambia, at the London School of Hygiene and Tropical Medicine (LSHTM). Sputum samples were sourced from parent studies which evaluated TB diagnostics in The Gambia [23,24]. All samples were from patients presenting with at least one or more symptoms concerning for TB, including: cough, weight loss, and shortness of breath. Samples derived from two different arms of the study (later referred to as Group A and Group B) which used different data collection tools to record clinical symptoms and histories. All participants were older than 18 at time of collection and resided permanently in The Gambia. Participants were recruited from the Greater Banjul area, The Gambia, which is the primary urban center in the country. Sputum samples were stored in an internal Biosafety Level-3 Biobank at −70°C after collection, with a backup diesel generator to guarantee ongoing power supply. Sputum positivity for TB had been previously confirmed using GeneXpert® by Cepheid (Sunnyvale, CA, USA). Freeze thaw cycles were not documented following their collection, however two cycles were assumed as these samples were only accessed for use in the parent studies followed by this investigation.

## Ethics approval and consent

Written informed consent for sample storage and further testing was obtained at the time of enrollment in the parent studies in all cases. Ethical approval for sample collection and storage was granted by the Gambia Government/MRC Joint Ethics Committee (Project ID/Ethics ref: 21727). Further ethical approval for screening of these samples for this study was granted by the same committee (Project ID/Ethics ref: 29250).

## Primer and probe design

The mitochondrial ribosomal small subunit RNA (mtSSU) gene (https://www.ncbi.nlm.nih.gov, accession number GG663449.1), internal transcribed spacer (ITS) (https://www.ncbi.nlm.nih.gov, accession number AF548061), and mitochondrial large-subunit ribosomal RNA (lsrRNA) gene (https://www.ncbi.nlm.nih.gov, accession number KC470812.1) were chosen as targets for *Histoplasma* spp., *Aspergillus* spp., and *Pneumocystis jirovecii*, respectively. Previously published primer-probe sets [25–27] with compatible cycling conditions were utilized, with minor augmentations to fit the experiment parameters. An internal control to detect human ribonuclease-P (RNaseP) from sputum was also included (https://www.ncbi.nlm.nih.gov, accession number NM_006413). Each primer-probe set was synthesized by IDT (San Diego, CA, USA) as a custom PrimeTime® real-time qPCR assay. Primer and probe sequences can be found in Table 1.

## *In silico* validation of primer and probe sequences

Primer-Blast® [28] was utilized to determine if any cross species-genus reactivity would be encountered for each respective primer probe pair. For *Histoplasma* spp. and *Pneumocystis jirovecii,* no significant homologies within the target area were detected. The same was encountered for the pan *Aspergillus* spp. target, except for *Penicillium spp.*. This cross-reactivity was already demonstrated in the primary publication, and the original authors did not note significant clinical impact [26].

## Positive controls

Due to lack of clinical samples in The Gambia containing DNA from target species, synthetic double stranded DNA positive controls at known concentrations were utilized. These were produced by IDT (San Diego, CA, USA) as gBlocks® gene

**Table 1. Primers and probe pairs.**

| Species | Target Gene | Primers/ Probes | 5' to 3' | Probe Dye (λ)/Quencher |
|---|---|---|---|---|
| *Histoplasma spp.* | mtSSU | Forward | CGTACGACATCATATTAAAAATGA | Cy5 (668); TAO/ Iowa Black RQ |
| | | Reverse | CTTTCTTTAAGGTAGCCAAAAT | |
| | | Probe | TGTAGTGGTGTACAGGTGAGT | |
| *Aspergillus spp.* | ITS | Forward | GTGGAGTGATTTGTCTGCTTAATTG | FAM (520); ZEN/ Iowa Black FQ |
| | | Reverse | TCTAAGGGCATCACAGACCTGTT | |
| | | Probe | CGGCCCTTAAATAGCCCGGTCCG | |
| *Pneumocystis jirovecii* | lsrRNA | Forward | CACTGAATATCTCGAGGGAGTATGAA | TET (539); ZEN/ Iowa Black FQ |
| | | Reverse | CTGTTTCCCTTTCGACTATCTACCTT | |
| | | Probe | TCGCACATAGTCTGATTAT | |
| Human RNase P | RNAse P | Forward | TGTTTGCAGATTTGGACCTGC | SUN (554); ZEN/ Iowa Black FQ |
| | | Reverse | AATAGCCAAGGTGAGCGGCT | |
| | | Probe | AAGGCTCTGCGCGGACTTGTGGA | |

Displays each primer and probe pair. Primer and probe sequences, in the order that they appear, were derived from [25–27] with some modifications to probe/quencher pairs.

fragments. Sequences of the synthetic genes produced for *Histoplasma* spp., *Aspergillus* spp., and *Pneumocystis jirovecii* can be found at https://www.ncbi.nlm.nih.gov, accession numbers MW317134.1 (position 1–355), MT649564.1 (position 1193–1390), and MH645509.1 (position 9–207).

### TB inactivation

Prior to extraction and qPCR, inactivation of potential TB in sputum samples was performed in a Biosafety Level 3 (BSL-3) laboratory. 200µl aliquots of sputum were heated at 95°C for 20 minutes and cooled under room temperature for 10 minutes prior to transferring to the extraction stage of the assay.

### Fungal DNA extraction

The extraction procedure was performed in a Biosafety Level 2 (BSL-2) laboratory. 200µl aliquots of sputum samples were pre-treated with a 10U/mL in PBS lyticase [29] solution for 30 minutes at 37°C for digestion of fungal cell wall. This was followed by 10 minutes of centrifugation at max speed (~21,000 g). For DNA extraction, the QIAMP® DNA mini kit (Venlo, Netherlands) was used following the manufacturer's guidelines. During the lysis stage, pellets were resuspended in Buffer ATL, Proteinase K, and Buffer AL respectively. For protein precipitation, ethanol (96–100%) was added followed by the removal of contaminants using the provided wash buffers. DNA was eluted in 50ul of the provided elution buffer. DNA concentration and purity for each sample was confirmed using a NanoDrop® (260nm/280nm~1.8).

### Multiplex assay

To avoid contamination, the qPCR assay was performed in three separate laboratories (master mix preparation laboratory, template addition laboratory, and thermocycler room). The following reagents were used for the multiplex assay: 2.5 µL of Universal Taqman® Master mix, 0.375 µL of each primer/probe mix to a final concentration of 500nM and 250nM respectively, and 1 µL DNA. 0.375µl of sterile water was added to adjust the reaction volume to 5 µL total. In singleplex reactions, the same final concentration of primer/probes were utilized, and reagents were adjusted to a total reaction volume of 5 µL.

Amplification was performed using the Biorad® CFX 96 (Hercules, CA, USA). Cycling conditions were 50°C for 2 minutes followed by 95°C for 10 minutes, then 39 cycles of 95°C for 15 seconds and 55°C for 1 minute.

A separate plate of samples was run concurrently on extracted DNA using the internal control Human RNaseP primers/probes to confirm presence of human DNA in the sputum specimen using the above reaction parameters.

### Limit of detection (LOD) and reaction efficiency

Serial dilutions by a factor of 10 of the respective positive controls at known concentrations were prepared ranging from $10^3$–$10^0$ copies/µL. Both singleplex and triplex reactions were performed to determine assay theoretical limit of detection (LOD) in both scenarios. Samples at each respective concentration were screened in triplicate. Dilution curves were utilized to determine reaction efficiency both for singleplex and multiplex reactions and to estimate any changes in efficiency when reactions were pooled. The multiplex assay sensitivity (LOD) was tested by running x10 replicate reactions containing all three primer and probe pairs, in three separate batches with single target positive controls at known concentrations near the theoretical LOD. This experiment was also utilized to validate multiplex specificity by ensuring no amplification in another target occurred.

### *In vitro* specificity

The assay was tested for *in vitro* cross reactivity on whole DNA extract from available species at University of California Los Angeles, USA, which included: *Candida albicans, Coccidioides immitis, Fusarium spp., Aspergillus niger, Aspergillus fumigatus,* and *Aspergillus terreus*.

## Precision

For intra-assay precision negative extraction control, positive control (PC), and no-template control (NTC) were tested on each plate. An invalid result was defined as a failed RNaseP internal control for a particular sample, after a second rerun of this sample. Any plates with failed positive controls or NTCs were rerun.

## Results

### Assay validation results

**qPCR efficiency.** The 10-fold dilution regression analysis under singleplex and multiplex conditions with respective genus or species are shown in Fig 1. Amplification was noted in all dilutions ranging from $10^3$–$10^0$ copies per microliter. Quantification Cycle (Cq) values were acceptable for each distinct target (*Histoplasma* spp.: 25.52–37.08; *Aspergillus* spp.: 25.14–35.78; *Pneumocystis jirovecii*: 27.41–38.97). Efficiencies were calculated at 95%, 110%, and 95% for the singleplex reactions and 80%, 99%, and 95% for the multiplex reactions, for the respective fungal species. A drop of 15% in reaction efficiency was noted for *Histoplasma* spp., but this was deemed to be acceptable given results of LOD experiments in the multiplexed reaction.

**Sensitivity (LOD) and specificity.** Using the 10-fold dilution experiment for the multiplexed reaction, we were able to estimate the theoretical Cq that correlated with the LOD for each target. Each assay showed 100% sensitivity at 10 copies per microliter. Additionally, a x10 replicate experiment was performed at near 5 copies per reaction, which demonstrated 100% sensitivity for *Aspergillus* spp. and *Pneumocystis jirovecii*, and 80% sensitivity for *Histoplasma* spp. Results can be found in Table 2. No non-specific amplification occurred in this experiment confirming that no interaction occurred between the respective primer-probe pairs. In the *in vitro* test of specificity, the *Histoplasma* spp. and *Pneumocystis jirovecii* targets did not amplify for *Candida albicans, Coccidioides immitis, Fusarium spp., Aspergillus niger, Aspergillus fumigatus,* and *Aspergillus terreus*. The *Aspergillus* spp. target did amplify for the *Aspergillus niger, Aspergillus fumigatus,* and *Aspergillus terreus* as expected and no cross-reactivity was observed for other clades.

**PCR cutoff values.** Amplification plots of all samples and from LOD experiments were assessed. No non-specific or concerning background amplification was noted below Cq of 40. This Cq was chosen as the cutoff for all targets.

### General demographics

273 sputum samples were screened by qPCR between February 2024 and June 2024. The sample population comprised 186 (68.1%) males and 87 (31.9%) females, with a median age of 34.5 years (range 18–70 years) recruited from a mix of ambulatory and inpatient settings. TB negative and positive status were 69.6% ($n = 190/273$) and 29.3% ($n = 80/273$) of participants, respectively, with 3 individuals with unknown TB status. A total of 13 individuals reported a history of previous TB infection ($n = 13/273$, 4.8%). As samples were derived from previous studies with limited data available, HIV status was recorded for only 205 (75.1%) samples, of which 15 individuals were HIV positive (7.3%). Table 3 summarizes the demographic data.

### Symptoms at the time of presentation

Questionnaire responses on symptoms at time of sputum collection were available for 205 (75.1%) individuals from two different arms (Group A and Group B) of the parent study. From Group A ($n = 151$), participants were asked about symptoms of cough, shortness of breath, weight changes, chest pain, and hemoptysis, with detailed responses summarized in Table 4. From Group B ($n = 54$) participants were asked about the presence of the following symptoms: cough, hemoptysis, fever, weight loss, fatigue, night sweats, chest pain, shortness of breath, nausea, low appetite, and weakness. Detailed responses are summarized in Table 5. Cough was the most common symptom reported by participants (Group A: $n = 151/151$, 100%; Group B: $n = 54/54$, 100%).

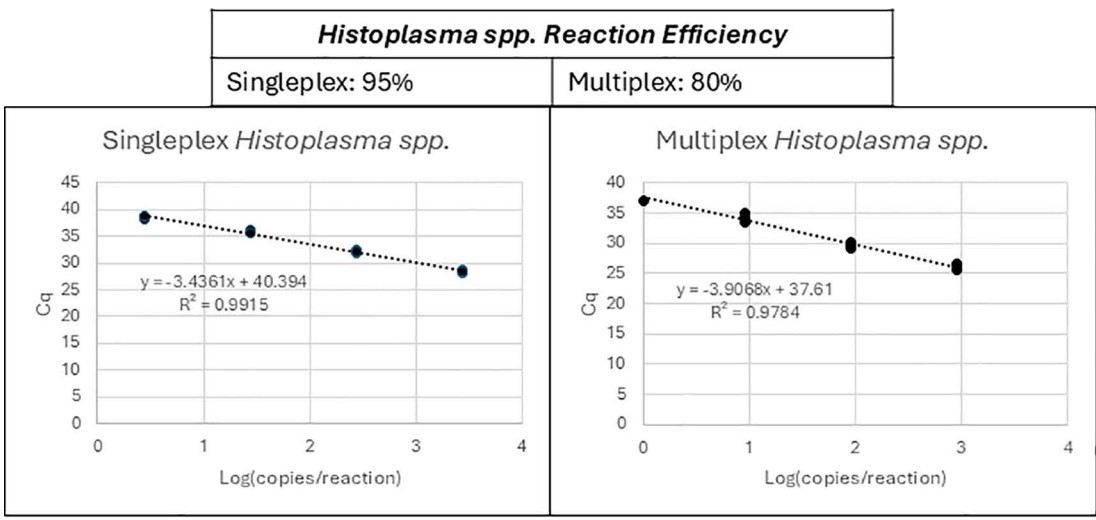

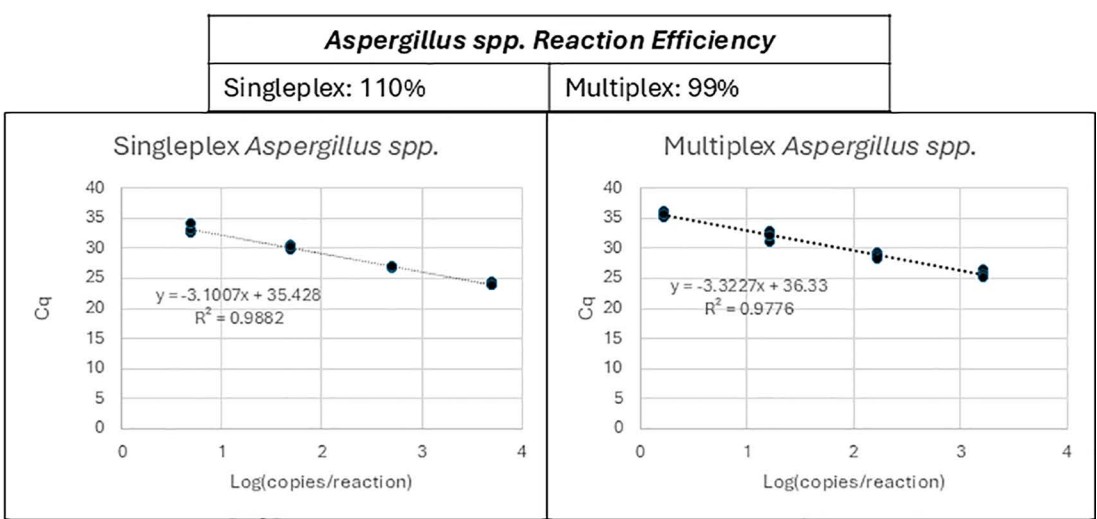

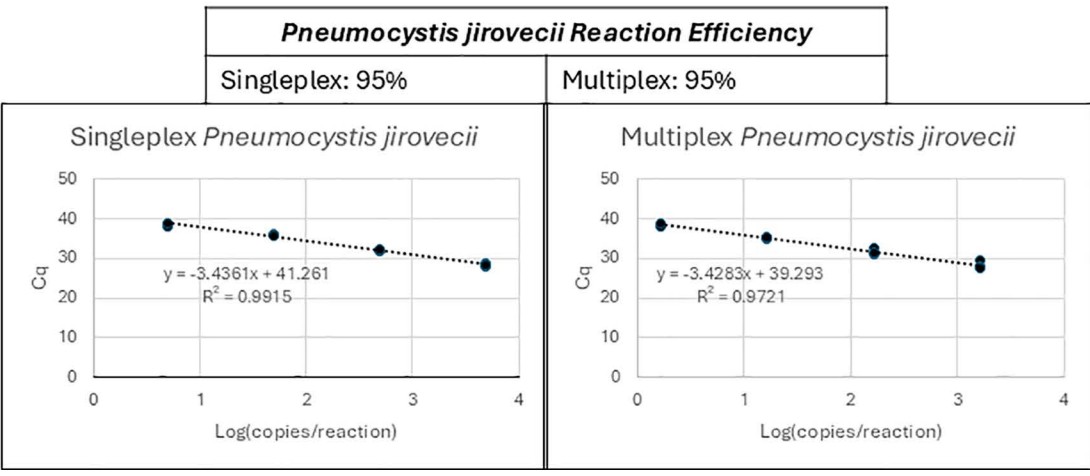

**Fig 1. Dilution regression analysis.** Positive controls at known concentrations ranging from 1000−10 copies per microliter were screened in both singleplex (single target) and multiplex reactions (pooled targets). Reaction efficiency in both scenarios as well as theoretical LOD was determined by

graphing the linear regression of the logarithmic transformation of the concentrations. Equations for the regressions are displayed in the figure with correlation coefficients (all were R²>0.95). (Cq = quantification cycle).

**Table 2. 10x Replicates at theoretical limit of detection.**

| Target: | *Aspergillus spp.* | *Histoplasma spp.* | *Pneumocystis jirovecii* |
|---|---|---|---|
| **Theoretical Cq at Limit of Detection:** | 35.8 | 37.6 | 38.8 |
| **Target concentration (copy/reaction):** | 5.33 | 3 | 5.44 |
| **Sensitivity at ~5 copy/ul:** | 100% | 80% | 100% |
| | Replicate Cq's | Replicate Cq's | Replicate Cq's |
| | 34.17 | 37.77 | 35.93 |
| | 34.5 | 35.3 | 38.72 |
| | 33.69 | 38.36 | 39.12 |
| | 33.22 | 37.96 | 38.56 |
| | 34.19 | 34.73 | 35.06 |
| | 33.29 | 35.27 | 36.44 |
| | 33.82 | No Amplification | 38.48 |
| | 33.38 | No Amplification | 38.03 |
| | 34.08 | 34.62 | 38.52 |
| | 34.2 | 35.03 | 36.2 |

Ten replicates at known concentrations near ~5 copies/uL were run on the multiplex platform. The average Cq at these concentrations was calculated as the theoretical Cq at limit of detection.

**Table 3. Basic demographic information.**

| Metric | Value |
|---|---|
| **Total Participants** | 273 |
| **Collection Date Range** | 10/02/2017–15/04/2021 |
| **Sex** | |
| *Males* | 186 (68.1%) |
| *Females* | 87 (31.9%) |
| **Age** | |
| *Average Age* | 35.98 years |
| *Age Range* | 18–70 years |
| *Median* | 34.50 |
| *Interquartile Range (IQR)* | 21.75 years |
| **Tuberculosis** | |
| *Tuberculosis Positive* | 80 (29.3%) |
| *Tuberculosis Negative* | 190 (69.6%) |
| *Unknown* | 3 (1.1%) |
| *History of Tuberculosis* | 13 (4.8%) |
| **HIV** | |
| *HIV Status Known* | 205/273 (75.1%) |
| *HIV Positive* | 15 (7.3%) |
| *HIV Negative* | 190 (92.7%) |

Basic demographic information of patients. Some demographic information may not have been available depending on inherent limitations from parent study which originally enrolled participants and collected the above data.

**Table 4. Group A symptoms.**

| Symptom | Participants | % |
|---|---|---|
| **Total Participants:** | **151** | |
| **Cough** | 151 | 100% |
| **Shortness of Breath** | | |
| *NYHA Class 1* | 28 | 18.5% |
| *NYHA Class 2* | 35 | 23.2% |
| *NYHA Class 3* | 2 | 1.3% |
| *NYHA Class 4* | 0 | 0.0% |
| **Weight Changes Over 3 Months** | | |
| *>10 kg* | 43 | 28.5% |
| *5–10 kg* | 68 | 45.0% |
| *No Change* | 0 | 0.0% |
| *Gained Weight* | 0 | 0.0% |
| *Do Not Know* | 3 | 2.0% |
| **Chest Pain** | | |
| *Daily* | 83 | 55.0% |
| *More than once per week* | 34 | 22.5% |
| *Less than once per week* | 7 | 4.6% |
| *Infrequently* | 3 | 2.0% |
| **Haemoptysis** | | |
| *Only specks* | 8 | 5.3% |
| *Less than 2 teaspoons* | 1 | 0.7% |
| *Between 2 teaspoons and 125 ml* | 2 | 1.3% |
| *>125ml* | 7 | 4.6% |

Breakdown of symptoms noted at time of sputum collection for group A. NYHA: (New York Heart Association).

**Table 5. Group B symptoms.**

| Symptoms: | Total | Mild | | Moderate | | Severe | |
|---|---|---|---|---|---|---|---|
| *Cough* | 54 | 16 | 29.6% | 37 | 68.5% | 1 | 1.9% |
| *Haemoptysis* | 7 | 6 | 85.7% | 0 | 0.0% | 1 | 14.3% |
| *Fever* | 48 | 20 | 41.7% | 26 | 54.2% | 2 | 4.2% |
| *Weight Loss* | 52 | 17 | 32.7% | 31 | 59.6% | 4 | 7.7% |
| *Fatigue* | 50 | 30 | 60.0% | 19 | 38.0% | 1 | 2.0% |
| *Night Sweats* | 43 | 20 | 46.5% | 23 | 53.5% | 0 | 0.0% |
| *Chest Pain* | 35 | 22 | 62.9% | 13 | 37.1% | 0 | 0.0% |
| *Shortness of Breath* | 28 | 21 | 75.0% | 7 | 25.0% | 0 | 0.0% |
| *Nausea* | 24 | 17 | 70.8% | 7 | 29.2% | 0 | 0.0% |
| *Low Appetite* | 45 | 21 | 46.7% | 24 | 53.3% | 0 | 0.0% |
| *Weakness* | 48 | 21 | 43.8% | 26 | 54.2% | 1 | 2.1% |
| Total number of samples with associated symptom data: 54 | | | | | | | |

Summary of symptoms noted at time of sputum collection for participants from group B.

## Positive samples

Of the 273 samples screened, 5 samples ($n = 5/273$, 1.8%) tested positive for the presence of *Aspergillus* spp. DNA, 3 ($n = 3/273$, 1.1%) tested positive for *Pneumocystis jirovecii*, and no samples tested positive for *Histoplasma* spp. Six samples were classified as indeterminant due to not amplifying the internal control or not meeting other quality control requirements. A summary of positive samples and associated demographic, clinical and symptom level data are summarized in Tables 6 and 7. For *Aspergillus* spp., cough and weight loss were the most common symptoms ($n = 4/4$, 100%), followed by chest pain ($n = 3/4$, 75%), and fever, shortness of breath, fatigue, low appetite, and weakness ($n = 2/4$, 50%, all symptoms). For *Pneumocystis jirovecii*, all patients ($n = 3/3$, 100%) endorsed cough, chest pain, and shortness of breath, two self-reported weight loss ($n = 2/3$, 66.6%), and one was HIV positive ($n = 1/3$, 33.3%). None of the positives for either fungal species reported a history of previous TB infection.

## Discussion

This study describes the development of a multiplex qPCR platform for the rapid screening and detection of *Aspergillus* spp., *Histoplasma* spp., and *Pneumocystis jirovecii* DNA in sputum by combining established primer-probe sets in a single reaction. This platform was utilized to screen for these three fungal targets in a sample of 273 patients with symptoms suggestive of TB in The Gambia. Fungal infections represent an underrecognized disease burden worldwide [1] and this work represents the first publication examining fungal DNA presence in sputum samples of adults in The Gambia. Using qPCR, 1.8% of sputum samples tested positive for *Aspergillus spp.* and 1.1% tested positive for *Pneumocystis jirovecii*. No sputum samples amplified *Histoplasma spp.* DNA.

This platform used three previously developed primer-probe sets, with some modifications, that were confirmed to be compatible via *in silico* analysis and had similar cycling parameters. No cross reactivity was found upon additional testing on synthetic double-stranded-DNA positive controls when tested on individual controls or pooled controls. We determined our protocol to have a sensitivity of 100% at 10 copies per microliter. To ensure future use of this assay at the MRC Unit, The Gambia, we developed a robust pipeline to safely inactivate sputum samples in a BSL-3 laboratory space followed by separate room extraction, preparation, and amplification steps. This allowed us to ensure both the safety of technicians given the high number of TB positive samples screened and lack of contamination between steps. This assay is now

**Table 6. Case-by-Case demographics.**

| Positive ID | Fungus identified | Age | Sex | Tuberculosis | HIV | Symptoms |
|---|---|---|---|---|---|---|
| 1 | *Aspergillus sp.* | 28 | F | Negative | Negative | *Cough, weight loss (5–10 kg), chest pain (>1/week)* |
| 2 | *Aspergillus sp.* | 20 | M | Negative | Negative | *Cough, SOB (NYHA Class 1), weight loss (>10 kg), chest pain (Daily)* |
| 3 | *Aspergillus sp.* | 30 | M | Positive | Negative | *Cough (moderate), fever (moderate), weight loss (moderate), fatigue (moderate), low appetite (moderate), weakness (moderate)* |
| 4 | *Aspergillus sp.* | 47 | F | Negative | Negative | *Cough (mild), fever (mild), weight loss (mild), fatigue (mild), chest pain (mild), SOB (mild), low appetite (mild), weakness (mild)* |
| 5 | *Aspergillus sp.* | 23 | M | Negative | No data | *No data* |
| 6 | *Pneumocystis jirovecii* | 61 | M | Positive | Negative | *Cough, SOB (NYHA Class 2), weight loss (5–10 kg), chest pain (<1/week)* |
| 7 | *Pneumocystis jirovecii* | 42 | M | Negative | Negative | *Cough, SOB (NYHA Class 2), weight loss (5–10 kg), chest pain (>1/week)* |
| 8 | *Pneumocystis jirovecii* | 29 | F | Negative | Positive | *Cough, SOB (NYHA Class 2), chest pain (>1/week)* |

This figure provides a case-by-case breakdown of symptoms at time of presentation for participants who tested positive.

**Table 7. Positive summary.**

| Data | *Aspergillus sp.* | *Pneumocystis jirovecii* |
|---|---|---|
| **Total** | 5 | 3 |
| **Age** | | |
| *Mean* | 31.9 | 44 |
| *Median* | 30 | 42 |
| *Range* | 20-47 | 29-61 |
| **Sex** | | |
| *Male* | 3 (60%) | 2 (66.6%) |
| *Female* | 2 (40%) | 1 (33.3%) |
| **Tuberculosis** | | |
| *Positive* | 0 | 1 (33.3%) |
| *Negative* | 5 | 2 (66.6%) |
| **HIV** | | |
| *Positive* | 0 | 1 (33.3%) |
| *Negative* | 4 | 2 (66.6%) |
| **Symptoms** | | |
| *Cough* | 4 (100%) | 3 (100%) |
| *Fever* | 2 (100%) | 0 |
| *Weight Loss* | 4 (100%) | 2 (66.6%) |
| *Fatigue* | 2 (50%) | 0 |
| *Chest Pain* | 3 (75%) | 3 (100%) |
| *Shortness of Breath* | 2 (50%) | 3 (100%) |
| *Low Appetite* | 2 (50%) | 0 |
| *Weakness* | 2 (50%) | 0 |

This figure summarizes the demographics of participants who tested positive.

available for both clinicians and other research scientists as part of a panel of assays at the MRC Unit, The Gambia, as both multiplex and singleplex species-specific assays. It is relatively cheap to utilize ($6.00 USD when taking into account cost of reagents and other consumables). Additional work is needed to compare its performance against other diagnostic modalities such as antigen- or culture-based methods in this resource limited setting. This work exemplifies the ease of implementation of a multiplex qPCR assay to screen patient samples for multiple targets concurrently, which has broad public health advantages.Overall, this pilot study uncovered a relatively low burden of fungal DNA, possibly representing colonization rather than infection, within the study population. Furthermore, the study population was relatively immuno-competent with only 15 individuals with known retroviral disease which represents a risk factor for all three groups of fungi [30]. Previous work demonstrated that 22% of symptomatic HIV patients in Africa tested positive for *P. jirovecii* via sputum PCR [21]. Worldwide, the incidence of *Aspergillus* spp. DNA present in the sputum varies considerably, with some studies estimating this number above 20% in patients with lower respiratory tract infections [31]. Recent work has demonstrated high rates of exposure to *Histoplasma spp.* in The Gambia [19]. Our study did not find any evidence of *Histoplasma spp.* within the sputum of participants. However, this could be due to several factors, including small sample size, location of infection (non-pulmonary presentations), and the relatively low number of HIV and TB positive patients. Future studies are needed to better assess presence of active human histoplasmosis in The Gambia.

This work found a relatively low rate of concomitant TB infection and fungal colonization versus infection. However, it should be noted that coinfection is not uncommon. For example, in other countries, the prevalence of concomitant TB and

Aspergillus infection is as high as 10%, particularly in cases of relapse post TB treatment [15]. Other work has noted the dangers associated with TB and *P. jirovecii* coinfection [32]. The low coinfection rate in our work may have been due to the small size of our study. Our results do show that colonization or infection from *Aspergillus* spp. is possible in The Gambia (and may occur at rates of near ~2% of patients presenting with symptoms concerning for TB), and clinicians should consider this diagnosis when faced with a clinical picture that fits the disease. Additionally, 1.1% of adults presenting with symptoms of TB infection were found to have *Pneumocystis jirovecii* DNA present in their sputum. Notably, one of these individuals was HIV positive. This both emphasizes the need for PJP prophylaxis in immunocompromised individuals in The Gambia, as well as the consideration of this infection in these patients presenting with acute respiratory symptoms. Typical presenting symptoms for both pulmonary aspergillosis and PJP overlap considerably, and include cough, fever, chest pain, weight loss, shortness of breath, and fatigue [20,33]. These symptoms were generally well represented in our patient cohort, including in patients with sputum containing fungal DNA. However, the low number of positive samples in our cohort made it unrealistic to evaluate for the statistical significance of these presenting symptoms, as well as age, sex, TB and HIV status and so descriptive statistics were reported.

## Limitations

Results from the sputum screening in this study should not be interpreted outside of the confirmed presence of *Aspergillus* spp. and *Pneumocystis jirovecii* DNA in the sputum of adults in The Gambia. This is secondary to several factors. Primarily, the limited size of the data set and confirmed positives restricts results interpretation. The study was not powered sufficiently given the number of positive samples to examine associations between a positive result and the demographic or clinical data collected. Importantly, previously collected samples were used from parent studies where fungal infection was not the original focus and this could have affected the results. Additionally, key risk factors, such as HIV infection, were not well represented in this cohort. qPCR, though relatively sensitive, is also well known to cause false positives for infection in the setting of sputum colonization [31]. This makes it difficult to interpret DNA amplification using this platform as clinically significant and requires clinicians to correlate this result with other clinical data such as chest x-ray or computed tomography. Future studies using other diagnostic modalities, combined with greater clinical details, are needed to better understand the use of qPCR for diagnosis of fungal infections in this setting. Additionally, though this assay was utilized as a screening tool in patients presenting with symptoms concerning for TB in this work, more positives may have been identified if patients had been screened for signs, symptoms, and radiographic evidence concerning for any of the three target species.

## Conclusion

In summary, we developed a multiplex qPCR assay for the detection of *Aspergillus* spp., *Histoplasma* spp., and *Pneumocystis jirovecii* from sputum in The Gambia. The assay performed well with acceptable sensitivity and specificity. It was used to screen 273 sputum samples from patients presenting with symptoms concerning for TB infection. *Aspergillus* spp. and *Pneumocystis jirovecii* DNA were found in 1.8% and 1.1% of these samples, respectively, and no samples tested positive for *Histoplasma spp*. This represents the first identification of *Aspergillus spp.* and *Pneumocystis jirovecii* in adults in The Gambia which supports clinician consideration of these two infections in patients presenting with symptoms suggestive of TB. Additional work is needed to better characterize fungal infections in The Gambia, to inform prioritization of resources for treatment of invasive fungal infections.

## Supporting information

**S1 Checklist. Inclusivity in global research questionnaire.**
(DOCX)

## Author contributions

**Conceptualization:** Zackary Salem-Bango, Tessa Rose Cornell, Jayne Sutherland, Karen Forrest, Omai Garner, Behzad Nadjm, Sheikh Jarju.

**Data curation:** Zackary Salem-Bango, Zainab Tambajang, Jayne Sutherland, Karen Forrest, Behzad Nadjm, Saffiatou Darboe, Basil Sambou, Sheikh Jarju.

**Formal analysis:** Zackary Salem-Bango, Sheikh Jarju.

**Funding acquisition:** Zackary Salem-Bango, Omai Garner.

**Investigation:** Zackary Salem-Bango, Mary Basiru Njai, Zainab Tambajang, Omai Garner, Saffiatou Darboe, Basil Sambou, Sheikh Jarju.

**Methodology:** Zackary Salem-Bango, Mary Basiru Njai, Zainab Tambajang, Tessa Rose Cornell, Omai Garner, Behzad Nadjm, Basil Sambou, Sheikh Jarju.

**Project administration:** Zackary Salem-Bango, Tessa Rose Cornell, Omai Garner, Sheikh Jarju.

**Resources:** Zackary Salem-Bango, Karen Forrest, Omai Garner, Behzad Nadjm, Saffiatou Darboe, Sheikh Jarju.

**Software:** Zackary Salem-Bango, Sheikh Jarju.

**Supervision:** Zackary Salem-Bango, Karen Forrest, Omai Garner, Sheikh Jarju.

**Validation:** Zackary Salem-Bango, Sheikh Jarju.

**Visualization:** Zackary Salem-Bango, Sheikh Jarju.

**Writing – original draft:** Zackary Salem-Bango, Mary Basiru Njai, Zainab Tambajang, Tessa Rose Cornell.

**Writing – review & editing:** Zackary Salem-Bango, Zainab Tambajang, Tessa Rose Cornell, Omai Garner, Behzad Nadjm, Sheikh Jarju.

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
