## [Decision Letter · Decision Letter 0]

11 Mar 2026

PONE-D-25-64712Prevalence of fungal infections in a patient cohort in The Gambia: Identification and characterization of three priority fungal species in patients with symptoms suggestive of TBPLOS One

Dear Dr. Salem-Bango,

Thank you for submitting your manuscript to PLOS ONE. After careful consideration, we feel that it has merit but does not fully meet PLOS ONE’s publication criteria as it currently stands. Therefore, we invite you to submit a revised version of the manuscript that addresses the points raised during the review process.

We look forward to receiving your revised manuscript.

Kind regards,

Rajeev Singh

Academic Editor

PLOS One

Journal Requirements:

https://journals.plos.org/plosone/s/file?id=wjVg/PLOSOne_formatting_sample_main_body.pdf and and and and https://journals.plos.org/plosone/s/file?id=ba62/PLOSOne_formatting_sample_title_authors_affiliations.pdf

3. In the online submission form, you indicated that your data is available only on request from a third party. Please note that your Data Availability Statement is currently missing [the name of the third party contact or institution / contact details for the third party, such as an email address or a link to where data requests can be made]. Please update your statement with the missing information.

Research reported in this publication was supported by the Global Health Program within the David Geffen School of Medicine at University of California Los Angeles

Reviewers' comments:

Reviewer's Responses to Questions

**Comments to the Author**

1. Is the manuscript technically sound, and do the data support the conclusions?

Reviewer #1: Yes

Reviewer #2: Yes

2. Has the statistical analysis been performed appropriately and rigorously? 

Reviewer #1: Yes

Reviewer #2: Yes

3. Have the authors made all data underlying the findings in their manuscript fully available?

The PLOS Data policy requires authors to make all data underlying the findings described in their manuscript fully available without restriction, with rare exception (please refer to the Data Availability Statement in the manuscript PDF file). The data should be provided as part of the manuscript or its supporting information, or deposited to a public repository. For example, in addition to summary statistics, the data points behind means, medians and variance measures should be available. If there are restrictions on publicly sharing data—e.g. participant privacy or use of data from a third party—those must be specified.requires authors to make all data underlying the findings described in their manuscript fully available without restriction, with rare exception (please refer to the Data Availability Statement in the manuscript PDF file). The data should be provided as part of the manuscript or its supporting information, or deposited to a public repository. For example, in addition to summary statistics, the data points behind means, medians and variance measures should be available. If there are restrictions on publicly sharing data—e.g. participant privacy or use of data from a third party—those must be specified.requires authors to make all data underlying the findings described in their manuscript fully available without restriction, with rare exception (please refer to the Data Availability Statement in the manuscript PDF file). The data should be provided as part of the manuscript or its supporting information, or deposited to a public repository. For example, in addition to summary statistics, the data points behind means, medians and variance measures should be available. If there are restrictions on publicly sharing data—e.g. participant privacy or use of data from a third party—those must be specified.requires authors to make all data underlying the findings described in their manuscript fully available without restriction, with rare exception (please refer to the Data Availability Statement in the manuscript PDF file). The data should be provided as part of the manuscript or its supporting information, or deposited to a public repository. For example, in addition to summary statistics, the data points behind means, medians and variance measures should be available. If there are restrictions on publicly sharing data—e.g. participant privacy or use of data from a third party—those must be specified.

Reviewer #1: Yes

Reviewer #2: Yes

4. Is the manuscript presented in an intelligible fashion and written in standard English?

Reviewer #1: Yes

Reviewer #2: Yes

5. Review Comments to the Author

Reviewer #1: This is a well written manuscript that describes very nicely the development and application of a multiplex PCR for fungal diseases in the Gambia.

The prevalence of fungal diseases in sputum collected from patients with TB suspcicion was low, probable due to the fact that they were not immunosuppressed.

Is there more information available on sociodemographic data? such as comorbidities, presence or not of cavitation in the chest xRay, socioeconomic data, housing conditions? What do you mean by TB status? Was TB confirmed for all samples? Were there ambulatory patients? hospitalized?

The discussion could be extended on the public health implications of the multiplex PCR. With these results, what would be the application of this tool? and how should it be used, in which group could it be applied for a better target of population at risk? Authors state that low coinfection rate in may have been due to the small size of our study. However it seems that the samples tested might have had a low pre-test probability for fungal infections- What is the info/literature available on who to test?

Reviewer #2: It is a good study. Would have been great to include a large sample size.

What will be the cost of this multiplex assay in low-resource setting? .

6. PLOS authors have the option to publish the peer review history of their article (what does this mean?). If published, this will include your full peer review and any attached files.). If published, this will include your full peer review and any attached files.). If published, this will include your full peer review and any attached files.). If published, this will include your full peer review and any attached files.

...

Reviewer #1: No

Reviewer #2: No

---

## [Author Response · Author response to Decision Letter 1]

30 Mar 2026

Dear Dr. Singh and Reviewers,

We sincerely thank you for your thoughtful evaluation of our manuscript entitled “Prevalence of fungal infections in a patient cohort in The Gambia: Identification and characterization of three priority fungal species in patients with symptoms suggestive of TB” and for the constructive comments provided. We appreciate the opportunity to revise our work and have carefully addressed each point raised. Below, we provide a detailed, point-by-point response to the reviewers’ comments and indicate where changes have been made in the revised manuscript.

Reviewer #1

We thank the reviewer for the positive assessment of our manuscript and for the helpful suggestions to strengthen the work.

Comment 1:

“Is there more information available on sociodemographic data? such as comorbidities, presence or not of cavitation in the chest xRay, socioeconomic data, housing conditions?”

Response:

We agree that additional sociodemographic and clinical data would enhance the interpretation of our findings. Unfortunately, detailed socioeconomic, clinical and housing data were not systematically collected as part of the parent data sets that our samples derived from—so we were not able to include this in our analysis and demographics figures.

Comment 2:

“What do you mean by TB status? Was TB confirmed for all samples? Were there ambulatory patients? hospitalized?”

Response:

We appreciate this request for clarification. We have revised the manuscript to clearly define “TB status” as sputum positivity via GeneXpert (Page 4, Lines 124-125). We have also clarified that patients were recruited from a mix of ambulatory and inpatient settings, though the data set we utilized did not specify which category each patient fell into (Page 8, Lines 236-238).

Comment 3:

The discussion could be extended on the public health implications of the multiplex PCR. With these results, what would be the application of this tool? and how should it be used, in which group could it be applied for a better target of population at risk? Authors state that low coinfection rate in may have been due to the small size of our study. However it seems that the samples tested might have had a low pre-test probability for fungal infections- What is the info/literature available on who to test?

Response:

Thank you for this important suggestion. We believe that this has been partially addressed in (Page 3, Lines 69-74). We have included additional discussion on (Page 9, Lines 293-294). Regarding which group this test could be utilized in, we have included an additional statement in (Page 10, Lines 336-338).

We based our sample size off of several previous works:

For Aspergillus sp., Lakoh et al demonstrated that 11.6% of patients presenting with respiratory symptoms had pulmonary aspergillosis in Sierra Leone. Granted, the HIV positivity rate in their sample population was ~50% which likely impacted the aspergillosis rate in comparison to our data set. This is discussed in (Page 10, Lines 303-305).

For Histoplasma sp., Cornell et al had demonstrated a high seroprevalence in The Gambia (~30%). In Nigeria, Ekeng et al had demonstrated that in patients with a presumptive TB diagnosis, ~13% tested positive for pulmonary histoplasmosis. 35% of their patients were HIV positive. This is discussed in (Page 9, Lines 300-301).

For Pneumocystis jirovecci, Wills et al showed a >20% prevalence of PJP pneumonia in patients with HIV in this region. This is discussed in (Page 9, Lines 298-299).

Unfortunately, due to this being a secondary study utilizing samples derived from parent works, we were not able to include as many samples from HIV positive individuals as we had originally hoped. We agree that testing patients with HIV, as well as TB like symptoms, would’ve likely led to higher positive results. We discuss this in (Page 10, Lines 330-331).

Reviewer #2

We thank the reviewer for their positive feedback and constructive suggestions.

Comment 1:

“Would have been great to include a large sample size.”

Response:

We agree that a larger sample size would strengthen the study. We discussed this in our “Limitations” section. Future work will reflect clinical implementation of this assay and we hope that data derived from this will allow for consideration of a larger group of patients.

Comment 2:

“What will be the cost of this multiplex assay in low-resource setting?”

Response:

Thank you for highlighting this important aspect. We have included an approximate cost of $6.00 USD per sample in (Page 9, Lines 291-292)

Additional Revisions and Journal Requirements

• We have updated the Data Availability Statement to include the required third-party contact details.

• We have added the Role of the Funder statement to our cover letter

• We have carefully reviewed and updated the reference list for completeness and accuracy.

• We have ensured that the manuscript complies with PLOS ONE formatting guidelines.

• We have completed and included the Inclusivity in Global Research questionnaire as Supporting Information.

• All changes in the manuscript are highlighted in the tracked changes version.

We believe that these revisions have significantly strengthened the manuscript and addressed all reviewer concerns. We are grateful for the reviewers’ insightful comments and the opportunity to improve our work.

Sincerely,

Dr. Zackary Salem-Bango and co-authors

---

## [Editor Report · Decision Letter 1]

7 Apr 2026

Prevalence of fungal infections in a patient cohort in The Gambia: Identification and characterization of three priority fungal species in patients with symptoms suggestive of TB

PONE-D-25-64712R1

Dear Dr. Salem-Bango,

We’re pleased to inform you that your manuscript has been judged scientifically suitable for publication and will be formally accepted for publication once it meets all outstanding technical requirements.

Kind regards,

Rajeev Singh

Academic Editor

PLOS One
---

## [Editor Report · Acceptance letter]

PONE-D-25-64712R1

PLOS One

Dear Dr. Salem-Bango,

I'm pleased to inform you that your manuscript has been deemed suitable for publication in PLOS One. Congratulations! Your manuscript is now being handed over to our production team.

Kind regards,

on behalf of

Dr. Rajeev Singh

Academic Editor

PLOS One